# A Cohort Study of the Influence of the 12-Component Modified Japanese Diet Index on Oral and Gut Microbiota in the Japanese General Population

**DOI:** 10.3390/nu16040524

**Published:** 2024-02-13

**Authors:** Satoshi Sato, Daisuke Chinda, Chikara Iino, Kaori Sawada, Tatsuya Mikami, Shigeyuki Nakaji, Hirotake Sakuraba, Shinsaku Fukuda

**Affiliations:** 1Department of Gastroenterology and Hematology, Graduate School of Medicine, Hirosaki University, Hirosaki 036-8562, Japan; satoshis@hirosaki-u.ac.jp (S.S.); chikaran0601@hirosaki-u.ac.jp (C.I.); hirotake@hirosaki-u.ac.jp (H.S.); sfukuda@hirosaki-u.ac.jp (S.F.); 2Division of Endoscopy, Hirosaki University Hospital, Hirosaki 036-8562, Japan; 3Center of Healthy Aging Innovation, Graduate School of Medicine, Hirosaki University, Hirosaki 036-8562, Japan; iwane@hirosaki-u.ac.jp (K.S.); tmika@hirosaki-u.ac.jp (T.M.); nakaji@hirosaki-u.ac.jp (S.N.)

**Keywords:** japanese dietary pattern, 12-component modified Japanese Diet Index, oral microbiota, gut microbiota, butyric acid-producing bacteria

## Abstract

The Japanese diet is a healthy dietary pattern, and the oral or gut microbiota have been identified as the main factors underlying the beneficial effects of the Japanese diet. However, epidemiological studies on Japanese dietary patterns calculated from daily eating habits in the general population yielded inconsistent findings. This study aimed to determine the association between the 12-component modified Japanese Diet Index (mJDI12) and the oral and gut microbiota in the general population of a rural area in Japan. After propensity-score matching, 396 participants (198 each in the low and high mJDI12 groups) were picked out. One year after the follow up survey, we reclassified the subjects and compared the low and high mJDI12 groups again. Participants with a high mJDI12 had a higher relative abundance of butyric acid-producing bacteria in their gut microbiota. Moreover, the significantly higher dietary fiber intake in the high mJDI12 group suggested that the high intake of dietary fiber contributed to an increase in butyric acid-producing bacteria in the gut. In contrast, in individuals with a high mJDI12, only *Allpprevotella* was decreased in the oral microbiota. Thus, the Japanese dietary pattern can have beneficial effects by improving the oral and gut microbiota.

## 1. Introduction

Diet has direct and indirect effects on host physiology through the microbiota [1,2]. Of the microbiota formed in various organs, oral and gut microbiota are greatly influenced by diet, and oral bacteria also influence the gut microbiota, which is called the oral gut microbiome axis [3,4].

The Japanese diet is characterized by a high intake of vegetables, soybeans, soybean foods, seaweed, mushrooms, fish and shellfish, green tea, and fermented foods [1,5]. The Japanese diet, along with the Mediterranean diet, is known to be a healthy dietary pattern and has been reported to be effective in preventing dementia and liver disease [6,7,8,9]. The health effects of the Japanese and Mediterranean diets are also mediated through the gut microbiota [10,11]. Moreover, the oral microbiota is also known to affect the development and progression of systemic diseases such as cardiovascular disease, diabetes, liver disease, and dementia, and the oral microbiota is also associated with dietary patterns [12,13,14,15,16].

The Japanese and Mediterranean diets are based on similar dietary patterns and include more vegetables, fish, and soybeans and less meat compared to Western diets [17,18]. Dietary fiber is classified as either water-soluble or insoluble based on solubility; soluble dietary fiber, which is abundant in vegetables, is a substrate for short-chain fatty acids such as butyric acid, propionic acid, and acetic acid and increases the levels of *Bifidobacterium*, *Lactobacillus*, and butyric acid-producing bacteria in the gut [19,20,21]. Fish containing polyunsaturated fatty acids increase *Bifidobacterium*, *Lactobacillus*, and *Akkermansia* in the gut and prevent obesity [22,23]. Soybeans contain soy isoflavones, a substrate for equol produced by the gut microbiota, which exhibits estrogenic activity [24].

Diet has also been reported to influence the oral microbiota [25,26]. The oral microbiota is strongly associated with the respiratory system but also influences the gut microbiota [3]. In patients of colorectal cancer and liver cirrhosis, major oral bacteria such as *Fusobacterium* and *Streptococcus* grow in the gut [27,28,29].

The 9-component Japanese Diet Index (JDI9) is a tool developed to assess traditional Japanese food patterns and scores the intake of 9 components (rice, miso, fish and shellfish, green and yellow vegetables, seaweed, pickles, green tea, beef and pork, and coffee) [8]. Subsequently, the 12-component modified Japanese Diet Index (mJDI12) was developed by adding 3 components (soybeans and soybean foods, fruits, and mushrooms) to the JDI9 [30]. A qualitative systematic review proved that the mJDI12 accurately assessed Japanese food patterns [5,30]. Because soybeans, soybean foods, and mushrooms are rich in dietary fiber, calcium, and magnesium, the nutrient density scores for these nutrients were better in the mJDI12 than in the JDI9 [30].

Although the various beneficial and harmful effects of the Japanese diet on the body have been reported to be mediated through the oral and gut microbiota, few studies dealt with daily eating habits and the Japanese dietary pattern. Furthermore, epidemiological studies have yielded different results depending on factors such as the region and age of the study population because the oral and gut microbiota are influenced by many confounding factors, including age, sex, body size, lifestyle habits, and medications [31].

We aimed to investigate the associations between Japanese dietary patterns, and the corresponding nutrients from daily food intake, and the oral or gut microbiota in the general population. We researched the changes in the influences of Japanese dietary patterns on oral or gut microbiota after adjusting for confounding factors.

## 2. Materials and Methods

### 2.1. Study Participants

This study was held as a part of the Iwaki Health Promotion Project. The Iwaki Health Promotion Project is a community-based health promotion project for Japanese residents that was designed to prevent the onset and progression of lifestyle-related diseases and prolong their lifespan. This project is carried out every June as a medical checkup for residents of the Iwaki region of Hirosaki City in the Aomori Prefecture, which is located in northern Japan [32]. All participants participated voluntarily, in response to a public announcement. In total, 811 adults between the ages of 19 and 87 participated in this project, which was conducted in June 2017 and June 2018 (Figure 1). Of these, 237 individuals with confounding factors that could affect their oral and gut microbiota, for example, malignant diseases, autoimmune diseases (inflammatory bowel disease, rheumatoid arthritis, autoimmune hepatitis, and Graves’ disease), diabetic mellitus, taking antibiotics, taking gastric acid secretion inhibitors, a history of gastric surgery, or those with missing data, were excluded. The remaining participants were classified into a low mJDI12 group (337 participants) and a high mJDI12 group (237 participants) based on the median mJDI12 score (6.0 points) at the time of the 2017 survey.

Furthermore, to equalize the background of both groups, propensity-score matching was carried out with adjustments for age, sex, and body mass index (BMI), which are confounding factors that cannot be eliminated even with exclusion criteria.

As a result of propensity-score matching, 396 participants, including 198 in the low mJDI12 group (Group L_1_) and 198 in the high mJDI12 group (Group H_1_), were detected, and a follow-up survey was carried out (Figure 1). One year later, in 2018, the two groups resorted to using the median mJDI12 score of 6.0 as the cut-off value for the low mJDI12 group (Group L_2_) and the high mJDI12 group (Group H_2_). Moreover, to further evaluate the impact of the mJDI12 on the oral and gut microbiota over time, the participants were stratified into four groups according to the change in the mJDI12 from 2017 to 2018: low to low mJDI12 group (L_1_-L_2_, 156 participants), high to low mJDI12 group (H_1_-L_2_, 65 participants), low to high mJDI12 group (L_1_-H_2_, 42 participants), and high to high mJDI12 group (H_1_-H_2_, 133 participants). Using an effect size of 0.25, a significance level of 5%, and a power of 95%, the required total sample size was calculated to be 280 cases. The number of subjects in this study was larger than the required sample size.

The bacterial diversity of the oral and gut microbiota and the relative abundance of bacterial species between the low and high mJDI12 groups in 2017 and 2018 were compared. The bacteria commonly observed in the high mJDI12 group in both 2017 and 2018 were defined as the bacteria related with mJDI12. Changes in the relative abundances of mJDI12-related bacterial from 2017 to 2018 were then investigated. Furthermore, the bacteria and nutrients associated with mJDI12 among the four groups at the time of the 2018 survey were compared.

### 2.2. Clinical Parameters

The mJDI12 was scored based on the Brief Self-administered Diet History Questionnaire (BDHQ), a convenient diet assessment questionnaire developed in Japan. The BDHQ is a self-administered questionnaire that evaluates the consumption frequency of foods to estimate the dietary intake of 58 commonly consumed foods and beverages over a period of one month [33]. The BDHQs were sent to participants in advance, and each participant was interviewed in detail and had their responses collected on the day of the project. We scored the mJDI12 based on the intake of soybeans and soybean foods, green and yellow vegetables, fruits, fish and shellfish, pickles, mushrooms, seaweeds, green tea, rice, miso soup, beef and pork, and coffee per 1000 kcal. We scored the participants as either at, below, or above the median intake of each food or food group separately for males and females [30]. For soybeans and soybean foods, green and yellow vegetables, fruit, fish and shellfish, pickles, mushrooms, seaweeds, green tea, rice, and miso soup, 1 point was added for an intake equal to or above the sex-specific median; for beef, pork, and coffee, 1 point was added for an intake below the sex-specific median.

The following parameters were investigated in the 2017 survey: age, sex, current and previous medical history, medications, height, and body weight. New diseases and medications observed between 2017 and 2018 were also recorded.

### 2.3. Next-Generation Sequence Analysis of Gut Microbiota

Saliva and fecal samples were collected in dedicated containers and suspended in a guanidine thiocyanate solution (100 mM Tris-HCl (pH 9.0), 40 mM Tris-ethylenediaminetetraacetic acid (EDTA; pH 8.0), and 4M guanidine thiocyanate) (TechnoSuruga Laboratory Co., Ltd., Shizuoka, Japan). The samples were kept at −80 °C prior to DNA extraction. A series of representative bacterial species in the human oral and gut microbiota were analyzed using primers for the V3–V4 region of the 16S rDNA of prokaryotes in accordance with previous studies [34]. Sequencing was carried out using the Illumina MiSeq system (Illumina, San Diego, CA, USA). The methods for quality filtering of the sequences were as follows: only reads with quality value scores ≥ 0 for more than 99% of the sequences were extracted for the analysis. Detection and identification of the bacteria from the sequences were performed using Metagenome@KIN software version 2.2.1 (World Fusion Co., Tokyo, Japan) and the TechnoSuruga Lab Microbial Identification database DB-BA 10.0 (TechnoSuruga Laboratory Co., Ltd., Shizuoka, Japan) at 97% sequence similarity. Relative abundance was presented as the percentage of reads for each bacterium relative to the total number of reads.

### 2.4. Statistical Analysis

The categorical variables were showed as frequencies, and the continuous variables as medians along with interquartile ranges. Chi-square and Mann–Whitney U tests were used to compare between the two groups. The Kruskal–Wallis test, followed by Steel–Dwass multiple comparisons, was used to compare among the four groups. Spearman’s rank correlation coefficients were used to investigate the correlation between changes in the bacterial species and intake of foods or food groups associated with mJDI12. The microbiota were compared using linear discriminant analysis effect size (LEfse) analyses [35].

Statistical analyses were performed using R software (R Foundation for Statistical Computing, version R-4.1.1) and Statistical Package for the Social Sciences (SPSS) version 28.0 (SPSS Inc., Chicago, IL, USA). Statistical significance was determined at *p* < 0.05.

### 2.5. Ethics Statement

Our study was conducted in accordance with the ethical standards of the Declaration of Helsinki and approved by the Hirosaki University Medical Ethics Committee (authorization numbers and ethical approval date were 2018-012, approved 11 May 2018, and 2021-030, approved 4 June 2021). We obtained informed consent from all the participants.

## 3. Results

### 3.1. Participant Characteristics

The baseline characteristics of the participants are presented in Table 1. The characteristics of Groups L_1_ (198 participants) and H_1_ (198 participants) after propensity-score matching for sex, age, and body mass index (BMI) are shown in Table 2. The two groups showed no significant differences in sex, age, or BMI. The median mJDI12 values at the time of the 2017 survey were 5.0 in Group L_1_ and 7.0 in Group H_1_.

Figure 2, Figure 3, Figure 4 and Figure 5 present differences in the composition and diversity of the oral and gut microbiota after propensity-score matching. There were no significant differences in the Chao-1 index, Shannon index, or principal coordinate analysis results for the oral microbiota (Figure 3). For the gut microbiota, the Shannon index in Group H_1_ was lower than that in Group L_1_, and the principal coordinate analysis revealed microbial differences in the gut microbiota between the groups (Figure 5).

### 3.2. Comparison of the mJDI12 and the Oral or Gut Bacterial Species in 2017 and 2018

The result of LEfSe for the mJDI12 and the oral or gut bacterial species after propensity-score matching are presented in Figure 6 and Figure 7. For oral species, only *Alloprevotella* was commonly decreased in group H in both 2017 and 2018 (Figure 6). In contrast, many species of bacteria were commonly observed in the gut in both 2017 and 2018 (Figure 7). In assessments of the commonly observed gut bacteria in both 2017 and 2018, the high mJDI12 group showed a decrease and increase in twelve and eight gut bacteria species, respectively (Figure 8). Of these, oral *Alloprevotella* and gut *Actinomycetaceae* and *Actinomyces* showed ˂1% relative abundance, while the other species showed a high relative abundance of >1%.

### 3.3. Participant Characteristics after Grouping Classified by the mJDI12 from 2017 to 2018

Participant characteristics and responses to the 2018 survey items, grouped according to the mJDI12 from 2017 to 2018, are presented in Table 3. Significant differences were found for age and foods and food groups other than rice (Table 3). Conversely, there were no significant differences among the four groups in both alpha and beta diversity results for both oral and gut microbiota (Appendix A). The amount of change in foods and food group intake from 2017 to 2018 is shown in Appendix A. There were slight changes for all groups and all foods and food groups from 2017 to 2018. No major changes were observed in the H_1_-L_2_ and L_1_-H_2_ groups, whose mJDI12 scores changed significantly over a year.

### 3.4. Correlation between Differences in the Intake of Foods and Food Groups and the Relative Abundance of Microbiota Associated with the mJDI12

Correlations between the amount of change in foods or food groups and the relative abundance of bacteria commonly observed in 2017 and 2018 survey are shown in Appendix A. There was no correlation for oral bacteria. In gut microbiota, a few bacteria showed a significant correlation, but the correlation coefficients were low and no consistent trends were observed in any of the four groups.

### 3.5. Association between Changes in the mJDI12 and the Relative Abundance of mJDI12-Related Bacteria

As of 2018, comparison of the microbiota associated with the mJDI12 at the genera level among the four groups showed that Group H_1_-H_2_ had a significantly higher relative abundance of gut butyric acid-producing bacteria, including *Feacalibacterium*, *Lachnospiracea_incertae_sedis*, and *Gemmiger*, than Group L_1_-L_2_ (Figure 9). In contrast, Group H_1_-H_2_ had a significantly lower relative abundance of oral *Alloprevotella*, gut *Bifidobacterium*, *Actinomyces*, and *Parabacteroides*.

### 3.6. Association between Changes in the mJDI12 and Nutrient Intake Levels

As of 2018, the comparison of nutrient intake and the mJDI12 showed that Group H_1_-H_2_ had a significantly higher intake of vegetable and animal proteins and soluble and insoluble dietary fiber than the other groups (Figure 10).

## 4. Discussion

To our knowledge, this is the first large cohort study to research the influence of mJDI12 on the oral and gut microbiota of the general population. Our study revealed that participants with a high mJDI12 had a higher relative abundance of butyric acid-producing bacteria, including *Feacalibacterium*, *Gemmiger*, *Ruminococcus*, and *Lachnospiracea_incertae_sedis*, and a lower relative abundance of *Bifidobacterium*, *Actinomyces*, and *Parabacteroides* in the gut. Moreover, only *Alloprevotella* of individuals with a high mJDI12 was decreased in the oral microbiota. The findings revealed that the intake of soluble and insoluble dietary fiber and vegetable and animal protein was high in group H_1_-H_2_, which showed a high mJDI12 in both 2017 and 2018.

This study revealed significant differences in the Shannon index, which reflected alpha diversity, and the principal coordinate analysis, which indicated beta diversity, between the low and high mJDI12 groups in the gut microbiota. A previous study reported that the group with high soluble dietary fiber intake showed lower alpha diversity and different beta diversity than the low-intake group [36]. Our results support the results of previous studies and suggest that the mJDI12 significantly influences the diversity of gut microbiota.

In this study, the bacteria commonly observed in both 2017 and 2018 were defined as species associated with high mJDI12 scores. Groups with high mJDI12 scores commonly showed an increase in butyric acid-producing bacteria, including *Feacalibacterium*, *Gemmiger*, *Ruminococcus*, and *Lachnospiraceae_incettae_sedis* in the gut. Furthermore, Group H_1_-H_2_ also showed a significantly higher relative abundance in *Feacalibacterium*, *Gemmiger*, and *Lachnospiraceae_incettae_sedis* in the gut than Group L_1_-L_2_ at the time of the 2018 survey. The high mJDI12 group showed a higher intake of soluble dietary fiber than the other groups. In particular, Group H_1_-H_2_ showed a significantly higher intake of soluble dietary fiber than Group L_1_-H_2_. High intake of dietary fiber, especially soluble dietary fiber, increases butyric acid-producing bacteria [36,37,38]. Most soluble dietary fibers reach the colon and are fermented into short-chain fatty acids, including butyric acid, propionic acid, and acetic acid. Butyric acid reduces intestinal permeability and inflammation through regulatory T cells, thereby reducing the inflow of toxic substances, such as endotoxins, into the liver [39,40]. Butyric acid can also prevent Alzheimer’s disease by reducing amyloid-beta levels in the brain and improving cognitive functions such as memory [41,42]. Furthermore, butyric acid can prevent the progression of metabolic dysfunction-associated steatotic liver disease by suppressing insulin-mediated fat accumulation and increasing anti-inflammatory effects via regulatory T cells [43,44]. Administration of butyric acid has been shown to strengthen skeletal muscle tissue [45]. *Feacalibacterium*, *Gemmiger*, *Ruminococcus*, and *Lachnospiraceae_incettae_sedis*, which showed higher abundance in the high mJDI12 group in this study, are butyrate-producing bacteria [46,47,48]. Of these bacteria, *Faecalibacterium* is the particularly abundant butyric acid-producing bacterium, and it is reduced in patients with inflammatory bowel diseases and metabolic dysfunction-associated steatotic liver disease [49,50,51]. Thus, the Japanese diet may prevent dementia and liver disease by increasing the number of butyric acid-producing bacteria. The high intake of soluble dietary fiber, a substrate of butyrate, could explain the increase in butyric acid-producing bacteria in the high mJDI12 group.

In this study, groups with high mJDI12 scores commonly had a lower relative abundance of *Bifidobacterium*, *Actinomyces*, and *Parabacteroides* in the gut. Furthermore, Group H_1_-H_2_ also showed a significantly lower relative abundance of *Bifidobacterium*, *Actinomyces*, and *Parabacteroides* in the gut than Group L_1_–L_2_ at the time of the 2018 survey. *Bifidobacterium* and *Actinomyces* belong to the *Actinobacteria* class, and *Bifidobacterium* has beneficial effects on the body and is used as a prebiotic. In contrast, *Actinomyces* is a major oral bacterium and is increased in the gut by dysbiosis [52]. *Parabacteroides* are influenced by diet, for instance, they are increased by the intake of sugar and monounsaturated fatty acids, and the increase in *Parabacteroides* is associated with obesity and memory loss [53,54].

In this study, the high mJDI12 group showed decreased numbers of gut *Bifidobacterium* strains in both 2017 and 2018. The results of our study were the opposite of those reporting that vegetables and fruits increase gut *Bifidobacterium* [20]. In the high mJDI12 group, coffee intake was significantly lower because 1 point was added for coffee intake below the median [30]. Coffee contains approximately 20% arabinogalactan and increases gut Bifidobacterium levels [55,56]. Additionally, *Bifidobacterium* levels decrease with age [57]. In fact, our study observed a significant negative correlation between *Bifidobacterium* and age (correlation coefficient: −0.209 in 2017 and −0.179 in 2018). Because most participants of this study were middle-aged or older adults, the association between the mJDI12 and *Bifidobacterium* strains differed from those in previous studies. The scoring method for coffee in the mJDI12 and the fact that the participants of this study were middle-aged and older may have influenced the results for the gut *Bifidobacterium* strains in this study.

This study revealed that, in the oral microbiota, only *Alloprevotella* was commonly decreased in group H in both 2017 and 2018. Furthermore, Group H_1_-H_2_ also presented a significantly lower relative abundance of oral *Alloprevotlla* in the gut than Group L_1_-L_2_ at the time of the 2018 survey. Oral *Alloprevotella* is known to increase with periodontitis [58]. The association between *Alloprevotella* and diet is unclear, but alcohol intake influences oral microbial structure and increases *Alloprevotella* [59]. Although the relative abundance of gut *Alloprevotella* is very low, gut *Alloprevotella* has harmful effects on the gastrointestinal tract and can be thought of as a potential oral biomarker for gastric diseases [60,61]. The relative abundance of gut *Alloprevotella* in this study’s participants was ˂0.01%, and there was no association between mJDI12 and gut *Alloprevotella*. However, this study suggested that the Japanese diet might have beneficial effects on the gut and the oral microbiota by reducing oral *Alloprevotella*.

The low and high mJDI12 score groups showed no significant differences in either alpha or beta diversity of the oral microbiota. Previous studies have reported that the diversity of oral microbiota differs between Koreans, who consume high amounts of spicy, salty, and fermented foods, and Japanese, who consume high amounts of vegetables, fish, and soybeans [26]. Another study comparing vegans and omnivores reported that vegans had higher alpha diversity and different beta diversities [25]. The results of this study, which showed no association between the mJDI12 and the diversity of the oral microbiota, differed from those of previous studies. Furthermore, in this study, only one oral species of *Alloprevotella* was commonly detected in both the 2017 and 2018 surveys. This study did not assess oral hygiene conditions, including the number of remaining teeth and periodontal disease, although it included mainly middle-aged and older adults, which could explain the differences in the results of the previous and present studies.

Several limitations exist in this study. First, the participants were middle-aged and older adults living in rural regions. Oral and gut microbiota are known to change with age and region; therefore, it is not appropriate to generalize our results to younger people or urban residents. Second, we did not assess oral hygiene conditions, including the number of remaining teeth and periodontal disease. Although this study did not reveal a significant association between mJDI12 and oral microbiota, unassessed oral hygiene may have influenced this result. Third, this study did not find a meaningful correlation between the intake of foods and food groups used to determine the mJDI12 and the relative abundance of oral and gut microbiota from 2017 to 2018 in any of the four groups. In this study, no major changes were observed in the H_1_-L_2_ and L_1_-H_2_ groups, whose mJDI12 scores changed significantly over a year. A longitudinal cohort study over a longer period than one year is necessary to resolve this limitation.

## 5. Conclusions

The Japanese dietary pattern changes the diversity of microbiota and increases butyric acid-producing bacteria in the gut. We also presented a decrease in oral *Alloprevotella*, which has harmful effects on both oral and gut microbiota; however, no significant differences were observed between the mJDI12 and the diversity of oral microbiota. This study suggested that the Japanese diet could have a beneficial effect on the host, with relationships observed between the Japanese diet and the oral and gut microbiota. Moreover, the mJDI12 is a useful scoring tool for assessing the oral and gut microbiota but is inadequate for assessing specific bacteria species such as Bifidobacterium strains; thus, there is room for improvement.

## Figures and Tables

**Figure 1 nutrients-16-00524-f001:**
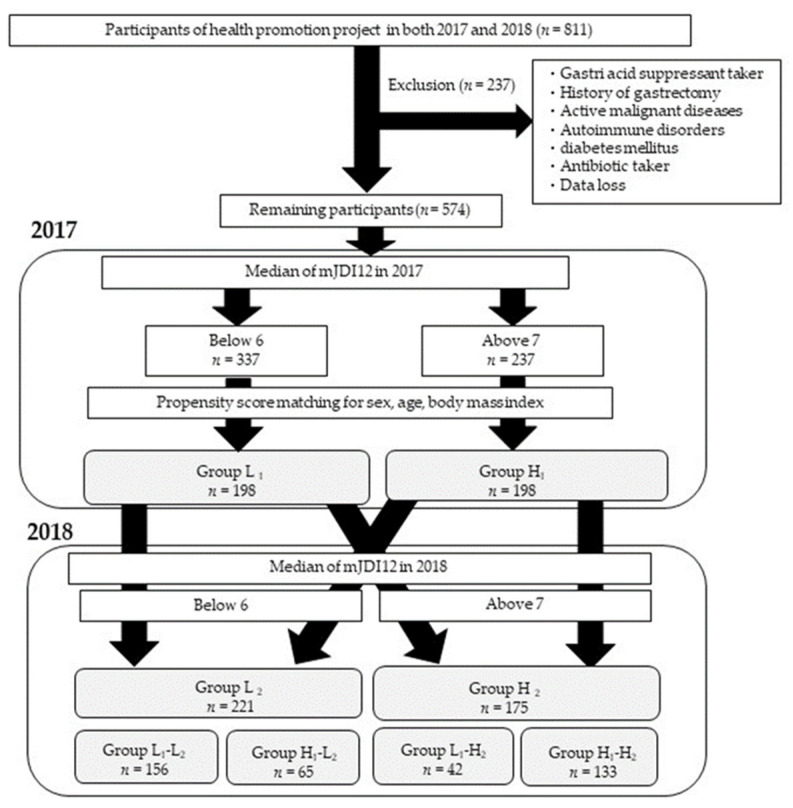
Study enrollment flowchart. mJDI12, the 12-component modified Japanese Diet Index. Group L_1_: mJDI12 ≤ 5 in 2017; Group H_1_: mJDI12 ≥ 6 in 2017; Group L_2_: mJDI12 ≤ 5 in 2018; Group H_2_: mJDI12 ≥ 6 in 2018. L_1_-L_2_: mJDI12 ≤ 6 in 2017 and 2018; H_1_-L_2_: mJDI12 ≥ 7 in 2017 and ≤ 6 in 2018; L_1_-H_2_: mJDI12 ≤ 6 in 2017 and mJDI12 ≥ 7 in 2018; H_1_-H_2_: mJDI12 ≥ 7 in 2017 and 2018.

**Figure 2 nutrients-16-00524-f002:**
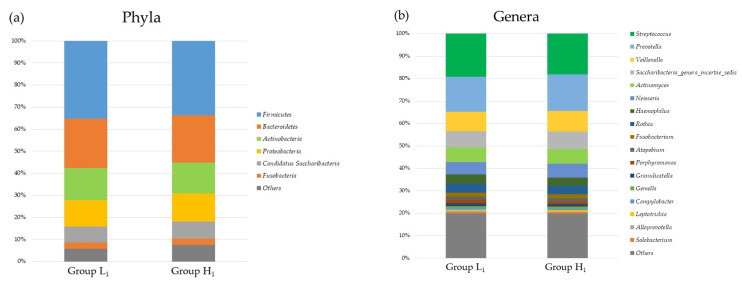
The composition of oral microbiota in the groups with mJDI12 ≤ 5 and ≥ 6 at (**a**) the phyla and (**b**) genera levels. Group L_1_: mJDI12 ≤ 6 in 2017. Group H_1_: mJDI12 ≥ 7 in 2017.

**Figure 3 nutrients-16-00524-f003:**
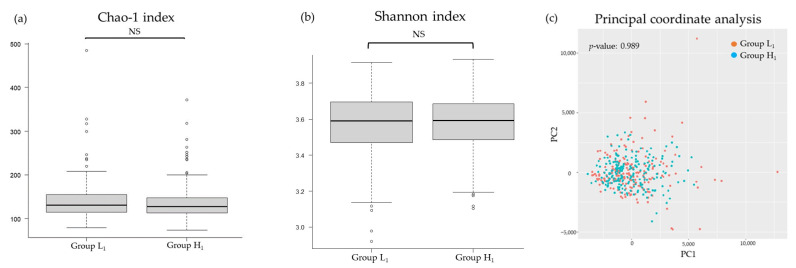
Differences in the diversity of oral microbiota in relation to the mJDI12: (**a**) Chao-1 index, (**b**) Shannon index, (**c**) principal coordinate analysis. Group L_1_: mJDI12 ≤ 6 in 2017. Group H_1_: mJDI12 ≥ 7 in 2017. NS, not significant; pc, principal components.

**Figure 4 nutrients-16-00524-f004:**
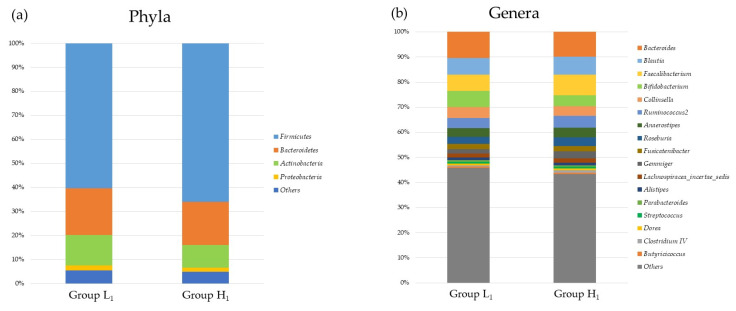
The composition of gut microbiota in the groups with mJDI12 ≤ 5 and ≥ 6 at (**a**) the phyla and (**b**) genera levels. Group L_1_: mJDI12 ≤ 6 in 2017. Group H_1_: mJDI12 ≥ 7 in 2017.

**Figure 5 nutrients-16-00524-f005:**
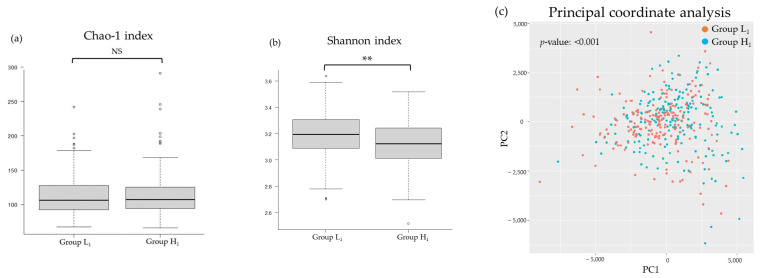
Comparison of the diversity of gut microbiota in relation to the mJDI12: (**a**) Chao−1 index, (**b**) Shannon index, (**c**) principal coordinate analysis. Group L_1_: mJDI12 ≤ 6 in 2017. Group H_1_: mJDI12 ≥ 7 in 2017. NS, not significant; pc, principal components. ** < 0.01.

**Figure 6 nutrients-16-00524-f006:**
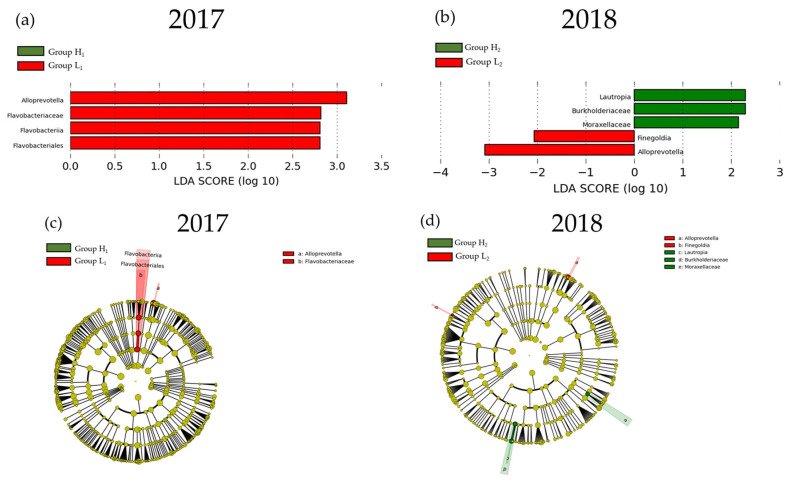
The LEfSe results of the oral microbiota in the low and high mJDI12 groups. (**a**) The linear discriminant for 2017. (**b**) The linear discriminant for 2018. (**c**) The cladogram report for 2017. (**d**) The cladogram report for 2018. Group L_1_: mJDI12 ≤ 6 in 2017. Group H_1_: mJDI12 ≥ 7 in 2017. Group L_2_: mJDI12 ≤ 6 in 2018. Group H_2_: mJDI12 ≥ 7 in 2018. LDA, linear discriminant analysis.

**Figure 7 nutrients-16-00524-f007:**
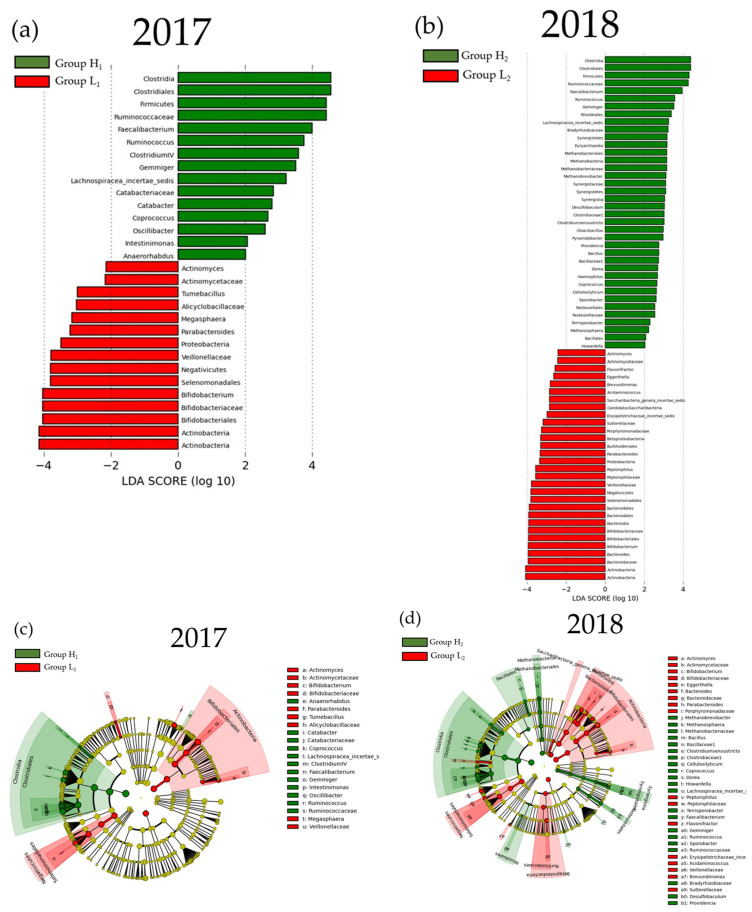
The LEfSe results of the gut microbiota in the low and high mJDI12 groups. (**a**) The linear discriminant for 2017. (**b**) The linear discriminant for 2018. (**c**) The cladogram report for 2017. (**d**) The cladogram report for 2018. Group L_1_: mJDI12 ≤ 6 of 2017. Group H_1_: mJDI12 ≥ 7 in 2017. Group L_2_: mJDI12 ≤ 6 in 2018. Group H_2_: mJDI12 ≥ 7 in 2018. LDA, linear discriminant analysis.

**Figure 8 nutrients-16-00524-f008:**
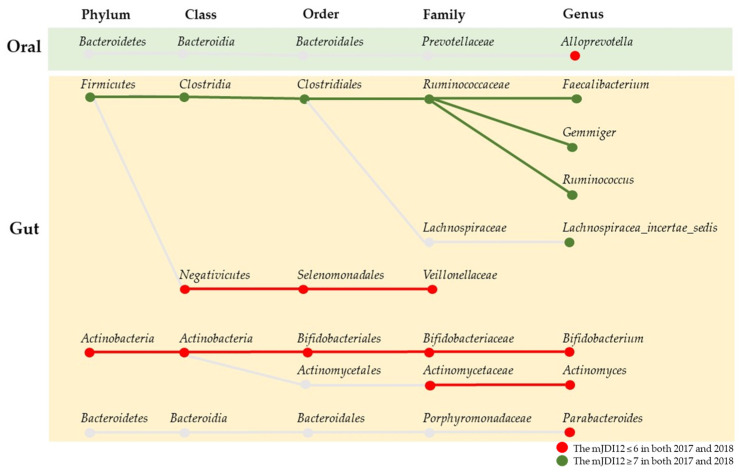
Systematic diagram of bacteria with significantly higher abundance in both years.

**Figure 9 nutrients-16-00524-f009:**
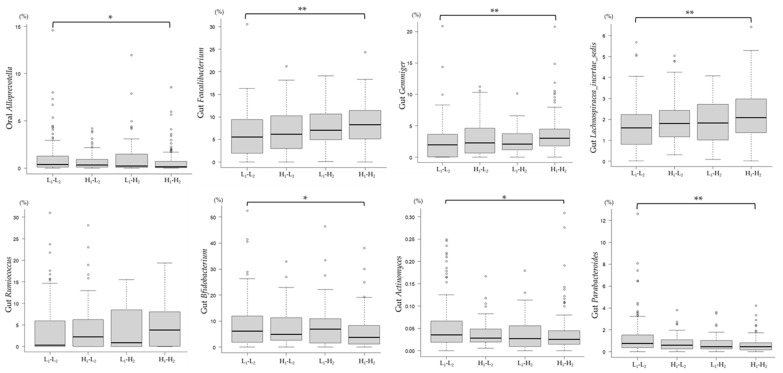
Genera-level comparison of gut bacteria among the four groups classified by the mJDI12 from 2017 to 2018. L_1_-L_2_: mJDI12 ≤ 6 in 2017 and 2018; H_1_-L_2_: mJDI12 ≥ 7 in 2017 and ≤ 6 in 2018; L_1_-H_2_: mJDI12 ≤ 6 in 2017 and mJDI12 ≥ 7 in 2018; and H_1_-H_2_: mJDI12 ≥ 7 in 2017 and 2018. * < 0.05, ** < 0.01.

**Figure 10 nutrients-16-00524-f010:**
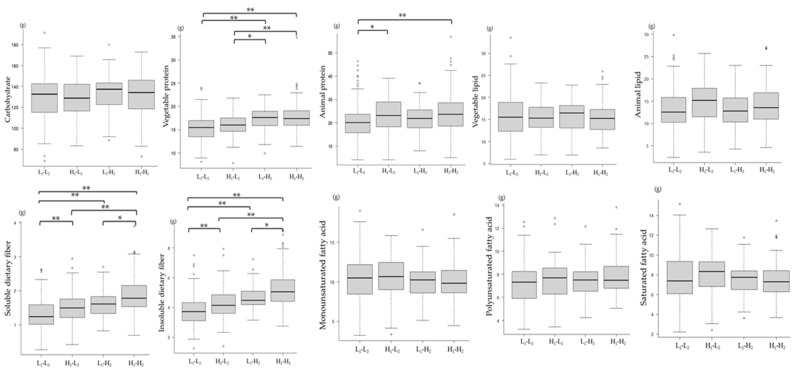
Comparison of the intake of nutrients among the four groups classified by the mJDI12 from 2017 to 2018. L_1_-L_2_: mJDI12 ≤ 6 in 2017 and 2018; H_1_-L_2_: mJDI12 ≥ 7 in 2017 and ≤ 6 in 2018; L_1_-H_2_: mJDI12 ≤ 6 in 2017 and mJDI12 ≥ 7 in 2018; H_1_-H_2_: mJDI12 ≥ 7 in 2017 and 2018. * < 0.05, ** < 0.01.

**Table 1 nutrients-16-00524-t001:** Participant characteristics at baseline.

Characteristics	mJDI12 ≤ 6 (*n* = 337)	mJDL12 ≥ 7 (*n* = 237)	*p*-Value
Sex (male/female)	139:198	103:134	0.607
Age (years)	49.0 (38.0–59.0)	60.0 (46.0–67.0)	<0.001
BMI (kg/m^2^)	22.5 (20.1–24.9)	22.5 (20.5–24.6)	0.672
mJDI12	5.0 (3.0–6.0)	8.0 (7.0–9.0)	<0.001
Soybeans and soybean foods	25.5 (15.1–39.3)	41.5 (28.6–56.2)	<0.001
Green and yellow vegetables	28.1 (17.8–44.1)	46.5 (34.3–65.8)	<0.001
Fruit	11.5 (4.6–23.7)	32.0 (16.3–55.5)	<0.001
Fish and shellfish	33.2 (22.1–47.9)	52.2 (37.8–68.3)	<0.001
Pickles	1.1 (0.0–3.9)	4.0 (0.9–8.4)	<0.001
Mushrooms	2.9 (1.7–6.0)	6.2 (4.0–10.3)	<0.001
Seaweeds	2.9 (1.6–6.5)	7.3 (5.6–12.0)	<0.001
Green tea	16.9 (1.7–65.1)	64.8 (23.3–175.0)	<0.001
Rice	167.1 (124.3–220.5)	178.5 (128.4–215.2)	0.666
Miso soup	62.9 (38.5–105.2)	102.5 (66.3–135.5)	<0.001
Beef and pork	19.1 (13.4–25.4)	16.2 (8.5–20.3)	<0.001
Coffee	110.5 (59.6–221.8)	83.1 (27.0–172.8)	<0.001

Number or median (range). BMI, body mass index.

**Table 2 nutrients-16-00524-t002:** Participants’ characteristics after matching for sex, age, and BMI.

Characteristics	Group L_1_ (*n* = 198)	Group H_1_ (*n* = 198)	*p*-Value
Sex (male/female)	85:113	79:119	0.610
Age (years)	56.0 (43.0–64.0)	57.0 (42.8–64.0)	0.771
BMI (kg/m^2^)	23.0 (20.5–24.8)	22.5 (20.5–24.6)	0.680
mJDI12	5.0 (3.8–6.0)	8.0 (7.0–9.0)	<0.001
Soybeans and soybean foods	24.7 (14.6–39.9)	41.6 (29.2–57.8)	<0.001
Green and yellow vegetables	30.2 (16.4–48.1)	46.4 (34.4–66.9)	<0.001
Fruit	13.3 (5.9–26.2)	30.6 (15.6–53.9)	<0.001
Fish and shellfish	33.8 (24.0–48.5)	50.9 (35.9–67.3)	<0.001
Pickles	1.6 (0.7–5.2)	3.5 (0.9–6.8)	0.004
Mushrooms	2.9 (1.6–6.9)	6.2 (4.4–10.1)	<0.001
Seaweeds	3.1 (1.6–6.9)	7.3 (5.6–11.6)	<0.001
Green tea	17.1 (5.1–56.6)	64.2 (15.4–170.7)	<0.001
Rice	152.9 (122.7–216.2)	178.1 (125.2–215.5)	0.310
Miso soup	64.8 (37.0–109.6)	104.2 (66.0–136.2)	<0.001
Beef and pork	19.0 (11.8–24.7)	16.3 (8.5–20.5)	0.002
Coffee	103.8 (56.2–214.8)	85.6 (40.5–176.9)	0.006

Number or median (range). Group L_1_: mJDI12 ≤ 6 in 2017. Group H_1_: mJDI12 ≥ 7 in 2017. BMI, body mass index.

**Table 3 nutrients-16-00524-t003:** Participants’ characteristics after grouping classified by the mJDI12 from 2017 to 2018.

Characteristics	L_1_-L_2_ (*n* = 156)	H_1_-L_2_ (*n* = 65)	L_1_-H_2_ (*n* = 42)	H_1_-H_2_ (*n* = 133)	*p*-Value
Sex (male/female)	71:85	27:38	14:28	52:81	0.474
Age (years)	54.5 (42.0–63.0)	49.0 (37.0–62.0)	62.0 (53.3–67.0)	59.0 (50.0–65.0)	<0.001
BMI (kg/m^2^)	22.6 (20.4–24.7)	22.9 (20.4–24.8)	23.2 (21.6–25.2)	22.4 (20.5–24.4)	0.560
mJDI12	5.0 (4.0–6.0)	6.0 (5.0–6.0)	7.0 (7.0–8.0)	8.0 (7.0–9.0)	<0.001
Soybeans and soybean foods	27.0 (15.1–37.5)	34.5 (20.3–51.6)	42.0 (32.0–51.9)	51.5 (35.8–62.5)	<0.001
Green and yellow vegetables	26.6 (15.4–44.3)	32.2 (19.6–49.3)	42.0 (29.1–59.4)	49.3 (32.8–74.9)	<0.001
Fruit	11.7 (4.0–26.0)	16.7 (7.9–45.7)	28.1 (16.9–48.0)	31.4 (19.2–56.6)	<0.001
Fish and shellfish	34.1 (22.9–51.5)	40.5 (24.9–63.4)	46.8 (32.1–58.5)	53.7 (38.3–68.9)	<0.001
Pickles	1.2 (0.0–4.9)	1.0 (0.0–4.4)	2.4 (0.8–4.9)	3.2 (0.9–7.5)	0.001
Mushrooms	3.4 (1.7–6.1)	4.6 (2.4–7.0)	5.4 (2.7–7.3)	6.8 (4.0–11.9)	<0.001
Seaweeds	2.9 (1.7–5.5)	4.6 (2.3–7.1)	7.4 (4.3–12.8)	7.6 (5.5–12.6)	<0.001
Green tea	15.5 (5.3–78.8)	24.8 (5.9–86.5)	41.8 (6.0–83.5)	70.7 (26.6–201.1)	<0.001
Rice	159.6 (104.5–208.8)	153.2 (126.1–182.5)	170.6 (129.5–207.0)	166.9 (126.8–213.3)	0.247
Miso soup	63.1 (36.8–96.5)	70.7 (43.4–118.1)	91.0 (67.5–125.1)	102.7 (61.8–144.4)	<0.001
Beef and pork	19.2 (11.7–26.4)	19.0 (12.8–28.9)	16.7 (7.4–20.3)	15.7 (8.8–20.9)	0.004
Coffee	129.5 (60.9–224.1)	139.4 (53.2–209.6)	96.7 (63.9–161.6)	82.4 (40.5–186.3)	0.014

Number or median (range). L_1_-L_2_: mJDI12 ≤ 6 in 2017 and 2018; H_1_-L_2_: mJDI12 ≥ 7 in 2017 and ≤ 6 in 2018; L_1_-H_2_: mJDI12 ≤ 6 in 2017 and mJDI12 ≥ 7 in 2018; H_1_-H_2_: mJDI12 ≥ 7 in 2017 and 2018.

## Data Availability

The data presented in this study are available upon request from the corresponding author. The data are not publicly available due to privacy and ethical restrictions.

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
