# Peer review of "A Cohort Study of the Influence of the 12-Component Modified Japanese Diet Index on Oral and Gut Microbiota in the Japanese General Population"

_nutrients, 2024, doi:10.3390/nu16040524_

Round 1

Reviewer 1 Report

Comments and Suggestions for Authors

Dear Authors,

your submitted manuscript contain important results for consumers. Well done!

I have only minor comments.

In your all manuscript change old no actual "microflora" to actual "microbiota"

figure 9 and 10 - should be biger or designed more clear

Line 384-385 - according your statement, So is it your next intention to continue studying these connections?

Author Response

Reviewer 1

We appreciate the time and effort that the reviewers have invested in evaluating our manuscript. We have revised the manuscript according to the reviewers’ suggestions, and our responses to each comment are listed below. The reviewers’ comments are presented in italics.

your submitted manuscript contain important results for consumers. Well done!

I have only minor comments.

  • In your all manuscript change old no actual "microflora" to actual "microbiota"

Response: Thank you for your comments. Per your suggestion, we have changed “microbiota.”

  • figure 9 and 10 - should be biger or designed more clear

Response: Thank you for your comments. As suggested, we have improved the resolution of Figures 9 and 10 to make them clearer.

  • Line 384-385 - according your statement, So is it your next intention to continue studying these connections?

Response: Thank you for your comments. Following comments from other reviewers, we reanalyzed the data and found an association between mJDI12 and oral Allpprevotella. Because the ‘Discussion section’ of the manuscript was modified, we have deleted the sentence (Lines 384-385) you commented on.

Reviewer 2 Report

Comments and Suggestions for Authors

The manuscript is based on an interesting concept – evaluating oral and gut microbiota in subjects following a certain type of diet. The narrative is unclear; authors should revisit the entire manuscript trying to present in a clearer way the design of the study and the results. Also, the link between the dynamics of gut microbiota and that of oral microbiota should be emphasized. In the present form, these two aspects seem to be two separate parts that do not fit.

The manuscript would benefit from literature citations from recent publications to support the introduction section as well as the discussion - DOI: 10.1146/annurev-physiol-031522-092054, DOI10.3390/nu15235005, doi: 10.1177/15353702231187645

Authors state : “The Japanese diet contains many antioxidant-rich ingredients, and their 37 antioxidant effects can prevent obesity, cardiovascular disease, and dyslipidemia” – obesity prevalence /incidence is surely not correlated with the antioxidant level in food, even if antioxidants can act at cellular level to prevent some complications of obesity. This should be corrected!

Authors state: ” The Japanese and Mediterranean diets are based on similar dietary patterns, and in-44 clude more vegetables, fish, and soybeans and less meat” – more/less compared to what???

 “Dietary fiber, which is abun-45 dant in vegetables, is a substrate for short-chain fatty acids such as butyric acid, propionic 46 acid, and acetic acid, and increases the levels of Bifidobacterium, Lactobacillus, and butyric 47 acid-producing bacteria in the gut [17-19].”- not all fibers generate short chain fatty acid, details should be included here

Authors tend to repeat info : “Dietary patterns have also been reported to influence the oral microbiota.”; “the oral microbiota is also associated with the dietary pattern”, etc. Authors should check the whole manuscript and eliminate repetitions

“Although the impact of the Japanese 280 diet on the diversity of gut microbiota has not been studied to date, foods and food groups 281 have been shown to have a significant impact on the diversity of gut microbiota [31,32].” – this is very general, a truism. This should be replaced

Authors state” In this study, the high mJDI12 group showed particularly high intake levels of soluble and 283 insoluble dietary fiber than the other nutrients.” – this is not a conclusion of the study, but a mere observation based on the type of diet the patients in the JDI12 group received. Authors should rephrase in order to focus in the discussion section only on conclusions based on their results as well as on correlations with literature data

“Administration of butyric acid has 304 been shown to strengthen skeletal muscle tissue and improve functional disability” ???

The stratification of participants presented in Study participants is unclear. This section should be re-written to better define the selection criteria as well as the relevance of these criteria. After that, these criteria should be followed for the section of the manuscript presenting the results and then in the discussion section.

The Conclusion section is too short and only repeats some info already included in the discussion section

Comments on the Quality of English Language

Language is generaly ok but some repetitions should be eliminated.

Author Response

Reviewer 2

We appreciate the time and effort that the reviewers have invested in evaluating our manuscript. We have revised the manuscript according to the reviewers’ suggestions, and our responses to each comment are listed below. The reviewers’ comments are presented in italics.

  • The manuscript is based on an interesting concept – evaluating oral and gut microbiota in subjects following a certain type of diet. The narrative is unclear; authors should revisit the entire manuscript trying to present in a clearer way the design of the study and the results. Also, the link between the dynamics of gut microbiota and that of oral microbiota should be emphasized. In the present form, these two aspects seem to be two separate parts that do not fit.

Response: Thank you for this suggestion. The other reviewer also commented that the study design should be modified. We have revised the entire manuscript to help readers better understand our study. Furthermore, we reanalyzed the data after excluding other confounding factors and found an association between the mJID12 and oral Alloprevotella. Based on this result, we emphasized the link between the gut and oral microbiota.

  • The manuscript would benefit from literature citations from recent publications to support the introduction section as well as the discussion - DOI: 10.1146/annurev-physiol-031522-092054, DOI10.3390/nu15235005, doi: 10.1177/15353702231187645

Response: Thank you for your informative articles related to our study. Based on your articles, we have revised the “Introduction and Discussion sections” to reflect the recent findings.

  • Authors state : “The Japanese diet contains many antioxidant-rich ingredients, and their 37 antioxidant effects can prevent obesity, cardiovascular disease, and dyslipidemia” – obesity prevalence /incidence is surely not correlated with the antioxidant level in food, even if antioxidants can act at cellular level to prevent some complications of obesity. This should be corrected!

Response: Thank you for your comments. As the reviewer has pointed out, this sentence was expressed too far in advance. We have deleted the sentence due to inappropriate text.

  • Authors state: ” The Japanese and Mediterranean diets are based on similar dietary patterns, and in-44 clude more vegetables, fish, and soybeans and less meat” – more/less compared to what???

Response: Thank you for your comments. As you pointed out, we did not list a comparison dietary pattern. We have added the sentence “compared to Western diets “and reference (Ogce F. Asian Pac J Cancer Prev. 2008 Apr-Jun;9(2):351-6.).

  • Dietary fiber, which is abun-45 dant in vegetables, is a substrate for short-chain fatty acids such as butyric acid, propionic 46 acid, and acetic acid, and increases the levels of Bifidobacterium, Lactobacillus, and butyric 47 acid-producing bacteria in the gut [17-19].”- not all fibers generate short chain fatty acid, details should be included here

Response: Thank you for your comments. As you pointed out, the fiber that generates short-chain fatty acids is water-soluble dietary fiber. We have corrected as follows ‘Dietary fiber is classified as water-soluble and insoluble based on its solubility in the gastrointestinal tract, and soluble dietary fiber, ・・・’

  • Authors tend to repeat info : “Dietary patterns have also been reported to influence the oral microbiota.”; “the oral microbiota is also associated with the dietary pattern”, etc. Authors should check the whole manuscript and eliminate repetitions

Response: Thank you for your comments. As you pointed out, we tended to repeat the same sentences. We have eliminated repetitions.

  • Although the impact of the Japanese 280 diet on the diversity of gut microbiota has not been studied to date, foods and food groups 281 have been shown to have a significant impact on the diversity of gut microbiota [31,32].” – this is very general, a truism. This should be replaced

Response: Thank you for your comments. As you pointed out, this sentence is very general, and the meaning is sufficiently understood without this sentence, so we have deleted the sentence.

  • Authors state” In this study, the high mJDI12 group showed particularly high intake levels of soluble and 283 insoluble dietary fiber than the other nutrients.” – this is not a conclusion of the study, but a mere observation based on the type of diet the patients in the JDI12 group received. Authors should rephrase in order to focus in the discussion section only on conclusions based on their results as well as on correlations with literature data

Response: Thank you for your comments. As you pointed out, this content is not a conclusion but a mere observation. We have deleted the sentence and revised the “Discussion section” to focus only on the conclusion based on our study result.

  • Administration of butyric acid has 304 been shown to strengthen skeletal muscle tissue and improve functional disability” ???

Response: We apologize for the inappropriate sentence. “improve functional disability” is unnecessary; therefore, we have deleted he sentence and corrected it as follows: ‘Administration of butyric acid has been shown to strengthen skeletal muscle tissue.’

  • The stratification of participants presented in Study participants is unclear. This section should be re-written to better define the selection criteria as well as the relevance of these criteria. After that, these criteria should be followed for the section of the manuscript presenting the results and then in the discussion section.

Response: Thank you for your comments. As you pointed out, the stratification of participants, including selection and exclusion criteria, seemed unclear. We stated at the beginning of the “Study participants” section that this study was conducted with general population volunteers in a health promotion project, and the participants were adults between the ages of 19 and 87. Moreover, we excluded the following confounding factors: active malignant diseases, autoimmune diseases (inflammatory bowel disease, rheumatoid arthritis, autoimmune hepatitis, and Graves’ disease), diabetic mellitus, and those taking antibiotics to exclude confounding factors related to this study. Furthermore, we added sample size calculations in the “Study participants” section to find the appropriate sample numbers for our study. Since the number of participants decreased from 623 to 574 in the first manuscript, the cut-off value of mJDI12 was changed from 5 to 6 points, and the subsequent results were revised accordingly. Furthermore, we added a new paragraph ‘3.3. Participant characteristics after grouping classified by the mJDI12 from 2017 to 2018.’ in the ‘Result’ section and modified the ‘Discussion’ section based on the revised results.

  • The Conclusion section is too short and only repeats some info already included in the discussion section

Response: Thank you for your comments. Per your suggestion, we have revised the ‘Conclusion’ section as follows: ‘The Japanese diet pattern changes the diversity of gut microbiota and increases gut butyric acid-producing bacteria. We also presented a decrease in oral Alloprevotella, which has harmful effects on both oral and gut; however, no significant differences were observed between the mJDI12 and the diversity of oral microbiota. This study suggested that the Japanese diet pattern could have a beneficial effect on the host with interrelationships between the oral and gut microbiota. Moreover, the mJDI12 is a useful scoring tool for assessing the oral and gut microbiota but is inadequate for assessing specific bacteria species such as Bifidobacterium strains, and there is room for improvement.’

Reviewer 3 Report

Comments and Suggestions for Authors

Sato et al. report on the role of Japanese diet in shaping gut microbiota based on data from 2018.

The introduction section is appropriately structured, although it could be somewhat shorter to keep focus of the reader.

Sample size for the study seems adequate, but this being a cohort study and considering the presence of 4 groups, the reader would benefit from information on sample size calculation based on the primary outcome of the study.

Considering its limitations, apart from the propensity score matching, can the authors perform comparison of relevant parameters (Chao-1 and Shannon index) on the original cohort but using multivariate adjustment for age, sex and BMI?

The exclusion criteria for do study do not seem to be adequate. There are many more possible confounders which affect gut microbiota: active malignant disease, DM, IBD, other autoimmune disorders... Thus, these participants should either excluded or accounted for. If such data is unavailable than the results cannot be interpreted at all.

Inclusion criteria are also quite elusive. 

Line 274 - minor error

Comments on the Quality of English Language

Minor English editing is needed

Author Response

Reviewer 3

We appreciate the time and effort that the reviewers have invested in evaluating our manuscript. We have revised the manuscript according to the reviewers’ suggestions, and our responses to each comment are listed below. The reviewers’ comments are presented in italics.

Sato et al. report on the role of Japanese diet in shaping gut microbiota based on data from 2018.

  • The introduction section is appropriately structured, although it could be somewhat shorter to keep focus of the reader.

Response: Thank you for your comments. Per your suggestion, we have shortened the ‘introduction section’ and kept the necessary information.

  • Sample size for the study seems adequate, but this being a cohort study and considering the presence of 4 groups, the reader would benefit from information on sample size calculation based on the primary outcome of the study.

Response: As the reviewer has pointed out, we calculated the sample size with an effect size of 0.25, a significant level of 5%, and a power of 95%. The required total sample size was calculated to be 280 cases. We have added this sentence in the ‘Study participants’  section.

  • Considering its limitations, apart from the propensity score matching, can the authors perform comparison of relevant parameters (Chao-1 and Shannon index) on the original cohort but using multivariate adjustment for age, sex and BMI?

Response: Following your comment, we compared the Chao-1 and Shannon index using multivariate analysis adjustment for age, sex, and BMI. The results of multiple analysis were the same as the results after propensity score matching, so only the results after propensity score matching are presented in the manuscript.

  • The exclusion criteria for do study do not seem to be adequate. There are many more possible confounders which affect gut microbiota: active malignant disease, DM, IBD, other autoimmune disorders... Thus, these participants should either excluded or accounted for. If such data is unavailable than the results cannot be interpreted at all.

Response: Thank you for your comments. As the reviewer has pointed out, the exclusion criteria were inadequate. We excluded the following confounding factors: active malignant diseases, autoimmune diseases (inflammatory bowel disease, rheumatoid arthritis, autoimmune hepatitis, and Graves’ disease), diabetic mellitus, and those taking antibiotics to exclude confounding factors related to this study. Since the number of participants decreased from 623 to 574 in the first manuscript, the cut-off value of mJDI12 was changed from 5 to 6 points, and the subsequent ‘Results’ and ‘Discussion’ sections were revised accordingly.

  • Inclusion criteria are also quite elusive.

Response: Thank you for your comments. Per your suggestion, we have added that the study participants are adults between the ages of 19 and 87 in the “Study participants’ section. Furthermore, we stated at the beginning of the “Study participants” section that this study was conducted with general population volunteers in a health promotion project.

  • Line 274 - minor error

Response: Thank you for pointing this out. We have modified the sentence.

Round 2

Reviewer 2 Report

Comments and Suggestions for Authors

Authors improved the manuscript based on the suggestion 

Comments on the Quality of English Language

Authors need to check again for language quality. Some examples below

Oral microbiota is strongly associated 57 with the pulmonary??? but also influences the gut microbiota [3].???

In 25 contrast, only Allpprevotella of individuals with a high mJDI12 was decreased in oral.???